

# The best of two worlds: reprojecting 2D image annotations onto 3D models

Marin Marcillat[1], Loic Van Audenhaege[1,2], Catherine Borremans[1], Aurélien Arnaubec[3] and Lenaick Menot[1]

[1] BEEP/LEP, Institut Français de Recherche pour l'Exploitation de la Mer, Plouzane, Bretagne, France
[2] Ocean Bio Geosciences, The National Oceanography Centre, Southampton, United Kingdom
[3] PRAO, Institut Français de Recherche pour l'Exploitation de la Mer, Toulon, Provence-Alpes-Côte d'Azur, France

## ABSTRACT

Imagery has become one of the main data sources for investigating seascape spatial patterns. This is particularly true in deep-sea environments, which are only accessible with underwater vehicles. On the one hand, using collaborative web-based tools and machine learning algorithms, biological and geological features can now be massively annotated on 2D images with the support of experts. On the other hand, geomorphometrics such as slope or rugosity derived from 3D models built with structure from motion (sfm) methodology can then be used to answer spatial distribution questions. However, precise georeferencing of 2D annotations on 3D models has proven challenging for deep-sea images, due to a large mismatch between navigation obtained from underwater vehicles and the reprojected navigation computed in the process of building 3D models. In addition, although 3D models can be directly annotated, the process becomes challenging due to the low resolution of textures and the large size of the models. In this article, we propose a streamlined, open-access processing pipeline to reproject 2D image annotations onto 3D models using ray tracing. Using four underwater image datasets, we assessed the accuracy of annotation reprojection on 3D models and achieved successful georeferencing to centimetric accuracy. The combination of photogrammetric 3D models and accurate 2D annotations would allow the construction of a 3D representation of the landscape and could provide new insights into understanding species microdistribution and biotic interactions.

## INTRODUCTION

Remote cameras, towed by research vessels or mounted on underwater platforms, have been used for decades for underwater exploration especially in the deep sea (*e.g.*, *Lonsdale, 1977*). Compared to physical sampling of the fauna, imaging is non-intrusive and non-destructive, and allows direct observation of the seabed over continuous areas (*Tunnicliffe, 1990*; *Beisiegel et al., 2017*). As a result, imaging has become a primary source of data to investigate interactions between seabed geomorphology and benthic megafaunal

Corresponding author
Marin Marcillat,
Marin.Marcillat@ifremer.fr

communities across spatial scales (*i.e.*, 10 s of m to kms). The method is particularly relevant for poorly accessible and/or vulnerable deep-sea ecosystems such as hydrothermal vents, cold seeps, canyons or coral reefs (*e.g.*, *Marcon et al., 2014*; *van den Beld, 2017*; *Robert et al., 2017*; *Girard et al., 2020*). Typical ecological investigations make use of geological and biological annotations from images (*Matabos et al., 2017*; *Schoening et al., 2017*). This annotation task is complicated by the fact that, in most cases, fauna cannot be identified down to the species level, making it susceptible to annotator bias (*Durden et al., 2016*). The recent development of image-based catalogues of fauna and seascape features (*e.g.*, *Althaus et al., 2015*; *Howell et al., 2019*) and the integration of these typologies into web-based annotation tools for 2D images has been widely used to mitigate identification bias by standardizing and remotely reviewing the categorization of large annotation sets (*e.g.*, *Langenkämper et al., 2017*).

For spatial investigation, image-based data needs to be located in a georeferenced system. In the deep sea, the location of the image is given by the navigation data of the submarine platform (*i.e.*, AUV, ROV or towed camera), which is provided relative to the position of the accompanying ship on the surface by a combination of dead reckoning and acoustic navigation. Dead reckoning means that the vehicle's position is calculated from its speed, heading and attitude usually provided by an inertial navigation system (INS) aided with a Doppler Velocity Log (DVL). To compensate for the drift of the inertial system and to provide a more accurate hybrid navigation, the dead reckoning navigation is periodically reset with the ship's position using an acoustic signal, typically the ultra short baseline system (USBL) (*Kwasnitschka et al., 2013*). While the ship gets its position from global navigation satellite systems with metric accuracy, USBL accuracy decreases with depth and distance. Depending on the system used and its calibration, the accuracy can range from 1% to 0.1% of the slant distance. At 1,000 m depth, the position accuracy is in the order of 10 m. The accuracy of image positioning may further be compromised by the horizontal distance between the transponder and the camera as well as the horizontal distance between the camera and the scene. The accumulation of inaccuracies in ship positioning, submarine platform positioning and scene positioning means that image-based data are seemingly inadequate to resolve abiotic and biotic processes operating at spatial scales lower than meters to decameters.

However, using recent advances in computer vision photogrammetry, overlapping images can be tightly aligned using a feature-matching algorithm while also refining *a posteriori* the position of the underwater camera. This allows the reconstruction of underwater scenes in three dimensions (*i.e.*, including overhang and cavities), advancing seascape ecology from 2.5D to 3D (*Kwasnitschka et al., 2013*; *Lepczyk et al., 2021*). Furthermore, the relative positioning of features over the resulting 3D model of the seabed can be as precise as 1 cm or even less (*Palmer et al., 2015*; *Istenič et al., 2019*). It should be noted however that while positions within a 3D model are internally consistent, the positioning of the 3D model itself still suffers from the inaccuracies of the navigation. But a 3D model also provides a digital terrain model (DTM) of a centimetric to millimetric precision, enabling a high-resolution mapping of the seabed bathymetry from which geomorphometric descriptors, such as slope and rugosity, can be rapidly and quantitatively

derived (*Wilson et al., 2007*; *Gerdes et al., 2019*). Those terrain metrics are especially of importance when considering them as driving ecological variables (*Robert et al., 2017*; *Price et al., 2019*).

A remaining problem, however, is that the georeferencing of the 3D model conflicts with the hybrid-based relocation of annotations made on the 2D images, hence producing two spatially misaligned datasets. A possible solution to cope with the mismatch between an hybrid-based positioning obtained from vehicle navigation and an optical-based positioning obtained from photogrammetry would still be to annotate the 3D model or the derived orthomosaics, instead of raw images. Some freely available software such as Potree (*Schütz, 2015*), 3DMetrics (*Arnaubec et al., 2023*) or commercial software such as VRGS (*Hodgetts et al., 2015*) or Agisoft Metashape (*AgiSoft, 2016*) already allow the direct annotation of 3D models. However, due to the additional reprojection step involved in their calculation, 3D textured models and associated orthomosaics typically have a slightly lower resolution than the raw 2D images and often present twirling artifacts, thus reducing the detectability of small organisms, and possibly, biasing the observed community composition (*Thornton et al., 2016*). While going back to the original image to identify the organism and then adding the annotation back to the 3D model is possible, this can take a significant amount of time. Annotations of 3D models can also be challenging due to the difficulty of displaying large high-resolution models and because of the longer duration required for drawing 3D geometries compared to 2D annotations. As a result, photogrammetry investigation typically focused on a subset of easily discernible organisms (*e.g.,* *Thornton et al., 2016*) or on areas of a few 10 s of m$^2$ (*e.g.,* *Lim et al., 2020*; *Mitchell & Harris, 2020*). Moreover, even if several 2D image annotation platforms exist (Bio-Image Indexing and Graphical Labelling Environment (BIIGLE): *Langenkämper et al., 2017*, Squidle+: *Bewley et al., 2015*, VARS: *Schlining & Stout, 2006*), equivalent collaborative software for 3D annotations are yet to emerge.

Because we identified the lack of open-access and open-source methods to mutualize the benefits of 2D image annotation on web-based annotation tools and photogrammetric outcomes (*i.e.,* internally accurate navigation and objective terrain descriptors), we propose an open-source workflow to project 2D annotations onto a georeferenced 3D model (*Marcillat et al., 2023*). This involves the development of a function that allows reprojection of annotations made in the open-access web-based image annotation tool BIIGLE (*Langenkämper et al., 2017*) onto 3D models produced with the freely available photogrammetry software Matisse3D (*Arnaubec et al., 2023*). A similar process had already been implemented in the commercial software Agisoft Metashape (*Pasumansky, 2020*), but our implementation is fully open-source and the entire workflow relies on open-access software. Here we explain the workflow and assess its accuracy in different deep seascapes, with two different submarine vehicles.

## MATERIALS AND METHODS

### Study sites

Four datasets were used to assess the accuracy of annotation reprojection onto 3D models (Table 1). The four datasets represented different geological settings acquired using two

**Table 1 Different datasets used during the reprojection error evaluation.**

| Site | Cruise | Location, latitude, longitude, depth | Topography | Acquisition |
|------|--------|--------------------------------------|------------|-------------|
| Tui Malila | Chubacarc (*Hourdez & Jollivet, 2019*) | Lau basin; −21.98905; −176.56844; 1,850 m | Highly complex seabed made of basalt concretion and collapse | • ROV Victor 6000<br>• Nikon D5500 |
| Eiffel Tower | Momarsat 2018 (*Cannat, 2018*) | Lucky strike field, mid-Atlantic ridge; 37.28951; −32.27500, 1,700 m | Vertical vent edifices surrounded by a mild-slope terrain | • 24 Mpx<br>• One frame/3 s |
| White Castle | | Lucky strike field; 37.29035; −32.28126; 1,700 m | | • 4 to 6 m altitude<br>• Automatic transects (survey mode) |
| Coral garden | ChEReef (*Menot & Tourolle, 2021*) | Lampaul Canyon, 47.61116; −7.53664; 800 m | Relatively flat terrace | • HROV Ariane<br>• Nikon D5500<br>• 24 Mpx<br>• 3 m altitude<br>• One frame/5 s<br>• Manual transect |

different vehicles. A downward-looking Nikon D5500 camera was mounted on the remotely operated vehicle (ROV) Victor 6000 to map three hydrothermal vent sites and on the hybrid remotely operated vehicle (HROV) Ariane to map a cold-water coral (CWC) reef.

On Victor and Ariane vehicles, underwater navigation is achieved through advanced sensor fusion techniques. Both vehicles employ a suite of similar equipment. A 600 kHz RDI DVL (Doppler Velocity Log) is utilized for precise velocity measurements and altitude estimation from the seabed. Absolute acoustic positioning is achieved using either the Posidonia 6000 or GAPS systems. Gyrofiber INS technology, such as the Phins from EXAIL, is employed to capture angle, angular velocity, and acceleration data. Depth measurements are obtained using Paroscientific sensors. These various navigation sensors are seamlessly integrated and processed by the INS Kalman filter, resulting in state-of-the-art acoustic/inertial navigation accuracy.

The underwater vehicles were operated at a constant altitude to acquire parallel photo transects to map the respective seafloor structures of the four sites. Tui Malila is a hydrothermal vent field surrounded by a complex basalt field and located on a fast spreading ridge in the center of the Lau back-arc basin (South-West Pacific; *Hourdez & Jollivet, 2019*). The modelled area is a rectangle of 250 by 10 m (Fig. 1C). Eiffel Tower and White Castle are two vent edifices on a gentle slope mainly made up of volcanic talus and are located in the Lucky Strike vent field on the Mid-Atlantic Ridge (*Ondréas et al., 2009*). At the periphery of the Eiffel Tower edifice, the modelled area is a rectangle of 120 by 10 m (Fig. 1D) and at the periphery of White Castle a rectangle of 115 by 30 m (Fig. 1B). The last dataset was acquired in a cold-water coral reef located on a large (150 by 50 m) and mostly flat terrace in a submarine canyon of the Bay of Biscay. The modelled area is a linear transect of 65 m long by 2.5 m width (Fig. 1A). The reef consists of isolated colonies of *Madrepora oculata* growing on a matrix of dead corals infilled with soft sediments.
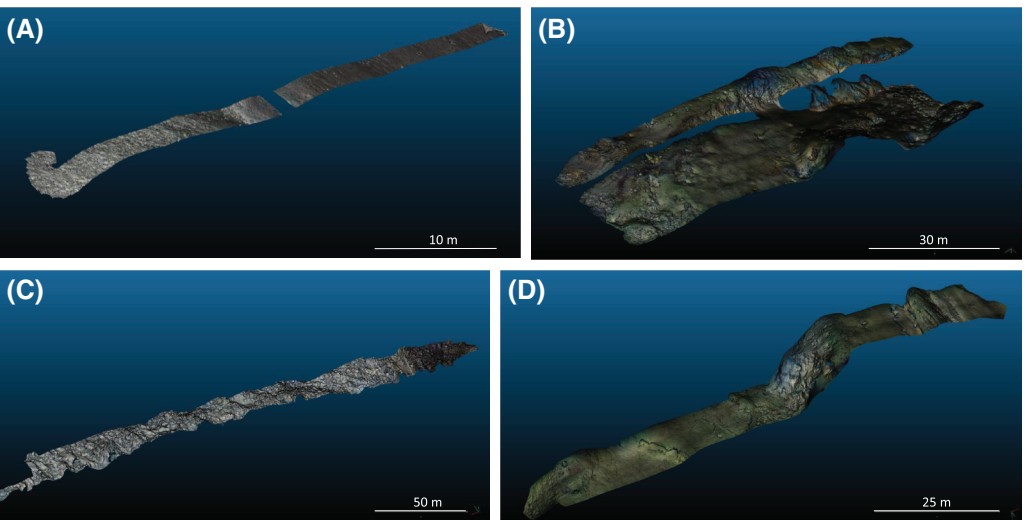

**Figure 1** **3D reconstructions used during this study.** (A) Coral Garden. (B) Periphery of White Castle vent site. (C) Tui Malila vent site. (D) Periphery of Eiffel Tower vent site.

## 3D reconstructions

For each of the four datasets, 3D models (*i.e.*, textured 3D meshes) were reconstructed in Matisse3D using the 3D Sparse FASTEST processing (*Arnaubec et al., 2023*). Prior to reconstruction and annotation, the images were corrected for underwater attenuation and non-uniform illumination using the Matisse3D preprocessing module in order to improve feature matching outcome. Images were also downscaled to 4 Mpx to speed up the reconstruction process.

Matisse3D performs feature detection and matching using the SIFT algorithm and removes outliers using the RANSAC model based on the fundamental matrix (*Arnaubec et al., 2015*). Bundle adjustment then uses the images to reconstruct the 3D points detected by the SIFT algorithm. During bundle adjustment, the position of the camera relative to the scene is modelled and georeferenced by minimizing the difference between these camera positions and those provided by the navigation system without altering the relative positions of the cameras in the bundle. The georeferenced camera positions resulting from the bundle adjustment are hereafter referred to as the optical navigation. The number of points in the model is then increased using the dense matching method in openMVS (*Cernea, 2020*) to create a dense cloud. Finally, a 3D mesh is generated using the Poisson surface reconstruction algorithm using the default values of the user parameters in openMVS and then textured using texrecon (*Waechter, Moehrle & Goesele, 2014*) with default user parameters. The resulting 3D models (textured mesh) have an average resolution of 5 mm.

## Image annotations

Two sets of image annotations were used for two different purposes. To build the first set, only images from the Tui Malila vent site were used. Disjoint images (images whose footprint do not overlap) were selected using the disjoint mosaic function of Matisse3D.

The function uses the hybrid navigation and the altitude of the ROV to map all images, then maximizes the number of theoretically non-overlapping images based on ROV-navigation. All these disjoint images were annotated for visible fauna (*i.e.*, on each image, all recognizable individuals were tagged with points, and all faunal patches were delimited with polygons) using the image annotation web service BIIGLE (*Langenkämper et al., 2017*). This disjoint-image annotation set was used to illustrate the mismatch in image positioning between the hybrid navigation and the optical navigation, and its consequences on annotation georeferencing (*e.g.*, double counting of some individual organisms and unnecessarily duplicating the annotation effort).

To build the second annotation set, a subset of 20 images was randomly selected from each of the four study sites and visually checked for non-overlap. On each image, four recognizable features were annotated with points using BIIGLE. These points were chosen carefully to ensure that they are evenly distributed throughout the image. Coordinates for all annotations in the image were then exported from BIIGLE in a format that provides individual positional information (*i.e.*, the CSV report scheme, see BIIGLE manual; *BIIGLE, 2024*). These points are hereafter referred to as "2D control points". This control dataset was used to assess the accuracy of the reprojection method (see below).

## Annotation reprojection

The images and/or the annotations from the disjoint and control datasets were reprojected onto the 3D models using the camera position and rotation information (the extrinsic parameters) and the optical characteristics of the camera (the intrinsic parameters) resulting from the photogrammetric reconstruction. This information is generated by OpenMVG (*Moulon et al., 2017*), which is the 3D reconstruction library implemented in Matisse3D. For the reprojection of each camera's line of sight, we used the Blender Python Library (BPL, *Blender Community Online, 2018*) and the Blender Photogrammetry Importer (*Bullinger, Bodensteiner & Arens, 2021*). In this process, the annotation features were then projected onto the 3D models using the ray tracing implementation in BPL (Fig. 2). For each of these features, a ray (aligned with the camera focal point and the image coordinates of the feature) is shot from the camera viewpoint towards the 3D model. The 3D reprojected position of a 2D feature corresponds to the first intersection between the ray and the model. If the reprojected annotation is a polygon, each vertex of the polygon is reprojected. If a ray does not hit the model, for example if the corresponding area in the image is not properly modelled, the annotation is discarded (Fig. 2). A Python implementation of this process is available from https://github.com/marinmarcillat/CHUBACAPP.

The camera corresponding to an annotated image is positioned and oriented in the 3D model referential and its optical characteristics are set according to the photogrammetric reconstruction result. For each annotation on the 2D image, a corresponding ray is shot towards the 3D model. Green: Successful point annotation reprojection. Blue: Successful polygon reprojection (each point of the polygon contour is reprojected). Orange: The ray missed the 3D model, the reprojection is unsuccessful.

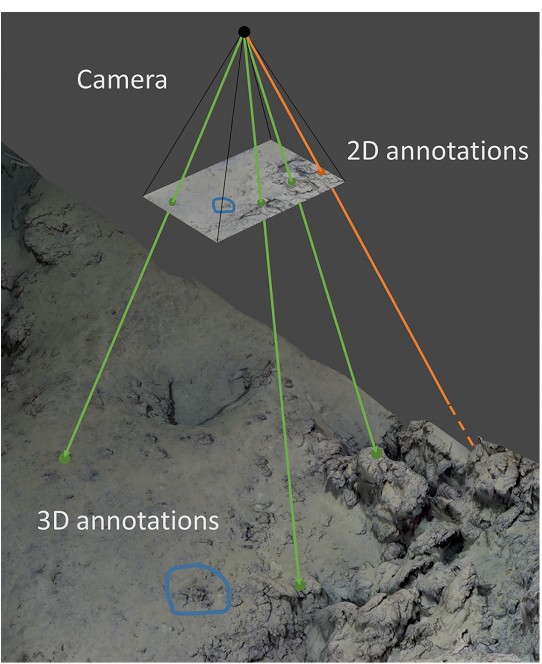

**Figure 2 Principle of annotation reprojection.** The camera corresponding to an annotated image is positioned and oriented in the 3D model referential and its optical characteristics are set according to the photogrammetric reconstruction result. For each annotation on the 2D image, a corresponding ray is shot towards the 3D model. Green: Successful point annotation reprojection. Blue: Successful polygon reprojection (each point of the polygon contour is reprojected). Orange: The ray missed the 3D model, the reprojection is unsuccessful.       

The local 2D uv coordinates of the features in the images were thus transformed into global coordinates (georeferenced using the WGS84 datum). The footprint of each image from the disjoint dataset was also determined as a 3D polygon by reprojecting the image corner coordinates in the same way as the annotation points.

## Evaluation of annotation duplicates and reprojection accuracy

The analysis of the disjoint image dataset and control dataset was conducted using the open-source visualization software Blender (*Blender Community Online, 2018*). In both cases, the corresponding textured 3D models were imported.

To determine the percentage of duplicates in the fauna annotations of the disjoint image dataset, these annotations were reprojected onto the 3D model and imported into Blender. When two reprojected annotations of the same species were in close proximity, the annotations on the original images were compared to assess whether they were of the same individual. The percentage of duplicated annotations was then calculated. To illustrate the areas of overlap, the footprint of each image (*i.e.*, a polygon corresponding to the entire image) was also reprojected onto the 3D model.

To evaluate the accuracy of reprojections, the 2D reprojected control points from the control dataset were imported into Blender. The features in the images corresponding to these 2D control points were visually identified within the textured 3D models. These points are hereafter referred to as 3D control points. The distance between 2D reprojected

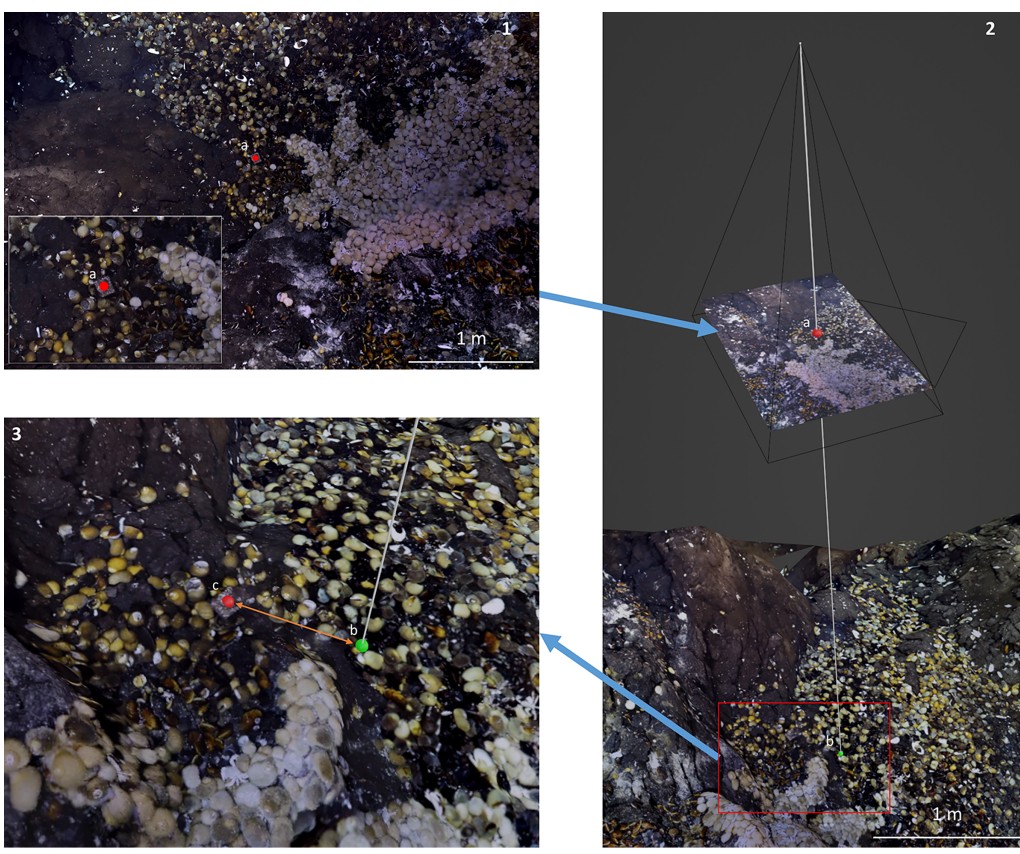

**Figure 3 Reprojection accuracy evaluation process.** (1) Annotation of a recognizable feature (2D control point, a) on the 2D raw image in BIIGLE (here, a cubic shape of the Tui Malila dive site). (2) Reprojection of this 2D control point onto the textured 3D model (green point, b). (3) The position of the feature annotated as a 2D control point is localized on the textured 3D model ("3D control point", red point, c), and the distance with the 2D reprojected control point (green point, b) determined.

control points and 3D control points was determined by comparing the 3D coordinates in Python using the Euclidean distance (Fig. 3). This 3D distance is preferred to the horizontal distance because in complex environments such as walls, the horizontal distance between two organisms on top of each other is nil and yet the distance along the surface is significant. The distance is used as a proxy for the error measurement of the annotation reprojection. For each dataset, the median Euclidean distance between 2D control points and 3D control points was computed together with the interquartile range (*i.e.*, IQR, the difference between the first and third quartile). The IQR is a measure of statistical dispersion.

## RESULTS

### Hybrid *versus* optical navigation

The discrepancy in image georeferencing between the hybrid and optical navigation is illustrated on Fig. 4 with the disjoint dataset. According to the hybrid navigation, images were aligned along three roughly parallel transects (Figs. 4A and 4B). According to the
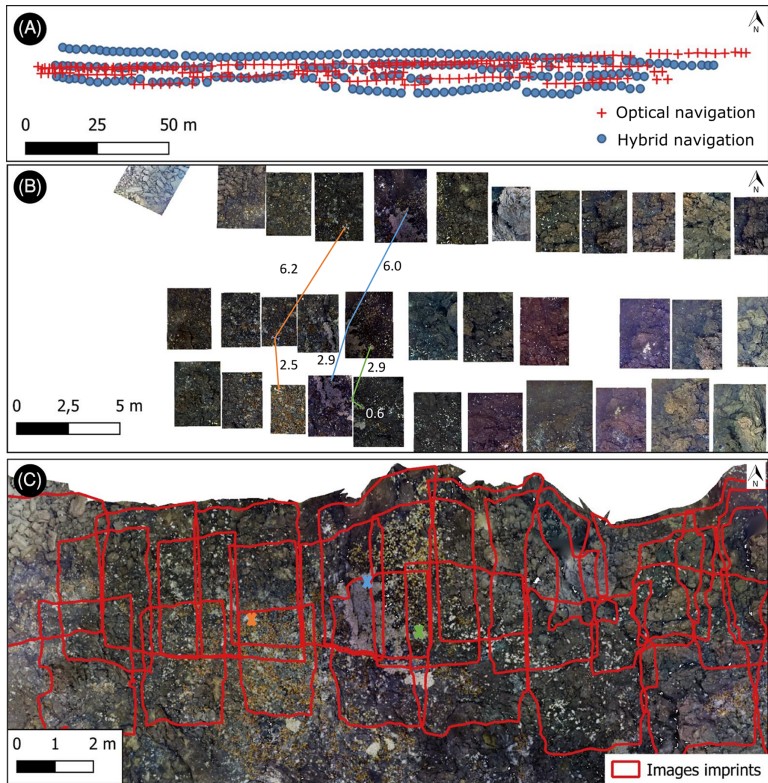

**Figure 4 Impact of navigational inaccuracies.** (A) Discrepancy between hybrid navigation and optical navigation (from photogrammetry). (B) 2D disjoint mosaic obtained with Matisse3D using hybrid navigation data. Three annotations, visible across several images, are shown separated by colored lines with the corresponding horizontal distance in meters. (C) Footprints of the same images reprojected onto the photogrammetric model. The duplicated annotations are represented with corresponding color crosses.

optical navigation however, the distance between the supposedly parallel transects decreases towards the west end (Fig. 4A). As a result, the footprint of images that were supposedly disjointed according to the hybrid navigation (Fig. 4B) actually overlapped on the textured 3D model, and 29% of the annotations were duplicated (Fig. 4C).

## Annotation reprojections

A total of 320 2D-control points were annotated on the raw images to assess the precision of reprojections, of which 293 could be successfully reprojected onto the 3D models. Points that failed to reproject mostly corresponded to unreconstructed areas (holes, sides) in the 3D models (17 missing points at Tui Malila, five at Eiffel Tower, three at the coral garden, and two at White Castle).

The distance between the reprojected 2D-control points and their reference position on the 3D models emphasize the accuracy achieved by reprojection (Fig. 5). The median distances at the coral garden (1.1 cm), White Castle (0.96 cm), and Eiffel Tower (1.3 cm) were similar, as well as their IQR, ranging from 1.2 cm at White Castle to 2.3 cm at Eiffel Tower. At Tui Malila, which is the topographically most complex site, the median distance and the IQR were higher, at respectively 8.6 and 21 cm.

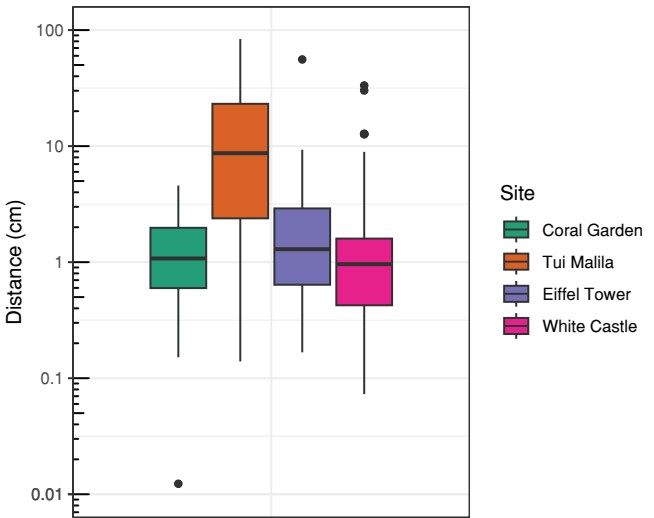

**Figure 5 Distribution of log-transformed distances between annotations of 2D control points on raw images and their corresponding 3D control points on 3D models for different sites.**

# DISCUSSION

The advent of underwater photogrammetry has offered a powerful tool to study seascapes, with a resolution hardly achievable by means of acoustic mapping of the deep seafloor. The same images acquired with an AUV or an ROV can now be used to map biological and geomorphological features as well as to build DTM with a resolution in the order of centimeters to millimeters. However, while there exist powerful open-access tools for image annotation and 3D model building, we found it difficult to merge data from these two processing pipelines because of their mismatch in georeferencing. We illustrated this discrepancy by reprojecting the footprint of supposedly disjoint images onto a 3D model. The images, and the annotations on these images, were georeferenced using the navigation of the ROV, as is usually done when processing such data. Even though images were georeferenced using a state-of-the-art navigation system combining acoustic and dead-reckoning positioning, the supposedly disjoint images were in fact overlapping on the 3D model, resulting in duplication of almost a third of the annotations made on these images. This is due to the well-known inaccuracy of even the most accurate positioning systems. The consequences are threefold. First, the absolute positioning of a feature is known within a radius that equals the accuracy of the positioning, that is to say in the range of 1 to 10 m. In the framework of seascape ecology, this may not be an issue if the accuracy is still higher than the resolution of the DTM on which the image data are mapped and against which they are analysed (*Swanborn et al., 2022*). But it becomes critical when the resolution of the DTM is an order of magnitude higher than the accuracy of annotation positioning. In addition, the relative distance between images is also approximate, which is limiting the application of spatial autocorrelation analyses. Finally, the same features may be annotated several times, which is a loss of time and can lead to an overestimation of relative abundance.

To cope with this discrepancy, we developed an open-access solution to reproject image annotations onto 3D models. Here, the position of the camera computed during the process of model building is used instead of the position of the camera given by the ROV navigation. In three of our study sites, image annotations, once reprojected, are positioned onto the 3D model with a median accuracy of about 1 cm compared to annotations made directly on the 3D model. Since the accuracy we achieved is similar to the resolution of the 3D model, the method allows to get the best of the two worlds, both annotations on images at full resolution with powerful annotation tools, and a true 3D DTM at very high resolution. Ultimately, this will facilitate exhaustive megafaunal community characterizations over large continuous spatial extents of 100 s of m$^2$, as those large image sets are increasingly collected with autonomous underwater vehicles (*Thornton et al., 2016*).

At the Tui Malila site however, the median accuracy was close to 10 cm and more variable than at the three other sites. Tui Malila is also topographically more complex than the other sites because of the rocky basaltic terrain exhibiting faults. Those faults could clearly limit the 'hit' of the ray traced from the downward-looking camera in near vertical setting, thus reprojecting the annotations a few decimeters away from the fault. The accuracy and precision of the reprojection are thus dependent on the roughness of the terrain, and further research is necessary to quantify this effect.

Terrain roughness also has an influence on the quality of the 3D model. In our datasets, 3% of image annotations could not be reprojected on 3D models due to model imperfections, most of which at the Tui Malila site. The ROV was pre-programmed to run along parallel paths to optimize the image overlap needed for 3D reconstruction. In this "survey" mode, the ROV heavily relies on the DVL for its navigation but in complex terrains with large bathymetric variations bottom tracking may be lost, thus compromising the integrity of the survey. New technological developments such as LIDAR, multibeam scanning sonars or stereo camera are now significantly improving the speed and the accuracy of 3D reconstruction and the overall quality of 3D mapping, as well as providing a simultaneous mapping and imaging of the seafloor at centimetric scales (*Caress et al., 2018*). These methods produce 3D models, bathymetry and orthophotomosaïcs that fully overlap, allowing the precise evaluation of species distribution in relation to topography (*Barry et al., 2023*). But while this mapping system requires specialized and costly equipment, reprojection can be utilized directly on most ROV, making it cost-effective.

A strong bottleneck that remains in image analysis is the time needed for annotations (*Matabos et al., 2017*). For image and video annotation, many online and collaborative tools have emerged (*e.g.*, BIIGLE: *Langenkämper et al., 2017*, Squidle+: *Bewley et al., 2015*, VARS: *Schlining & Stout, 2006*), and the latest developments in assisted feature annotation have been integrated (*Zurowietz et al., 2018*). Citizen science platforms (Deep sea spy: *Matabos et al., 2018*, Zooniverse: *Simpson, Page & De Roure, 2014*) also allow a significant increase in the amount of images processed. Our ability to adapt the workflow to BIIGLE demonstrates that in principle it could also be adapted to the annotation platforms mentioned above. Reprojection could also be useful for disjoint image selection. Optically overlapping images can be detected before annotation by reprojecting the image footprints

and checking for overlap between these reprojections. Furthermore, automated detection by machine learning looks very promising to speed up the process. Although some experiments on automatic recognition of 3D features have been carried out (*De Oliveira et al., 2021*), most detection models remain actually developed for 2D images and videos (*Katija et al., 2022*). Reprojection may allow the use of well-proven generic 2D image detection convolutional neural networks (CNN) for 3D annotation generation and vice-versa. On the one hand, 3D reprojection positions could generate 3D annotation sets for machine learning training. On the other hand, once a feature has been manually annotated on a single image and reprojected onto the 3D model, that 3D position can be used to locate that feature back on images taken from different angles. Multiple crops of the same object could then be generated and serve as a training dataset.

Ultimately, fully 3D open-access and collaborative annotation of high-resolution textured 3D models appear as an ideal solution in terms of accuracy for the analysis of species microdistribution and biotic interactions. This is already possible with some commercial solutions such as VRGS (*Hodgetts et al., 2015*). However, the cost of manipulating 3D models in terms of computational and power requirements must be taken into account. Large 3D models require significant amounts of memory and graphics processing power to manipulate, which may require dedicated computers and/or servers. By removing the need for 3D annotation and thus reducing the number of 3D models manipulations, our solution could provide a trade-off between accuracy and computational requirements.

## CONCLUSIONS

This study demonstrated that underwater photogrammetry and 2D annotations projected onto a 3D seascape model have several advantages when combined, in particular:

- Instead of working on orthophotomosaics of lower quality, the raw seascape images can be annotated using the popular and collaborative BIIGLE software at full resolution, allowing optimal morphospecies categorization.
- The positional accuracy of the 3D projection of the 2D annotations is compatible with the analysis of intraspecific and interspecific spatial interactions.
- The 3D model provides access to high-resolution topographic metrics to explain organism distribution over several spatial scales.
- By removing the need of annotating and manipulating 3D models, this workflow (using our open-source tool Chubacapp or commercial tools such as Metashape) could be applied to very large mapping areas, including particularly complex terrains such as hydrothermal vents, canyons, cliffs, or coral reefs. We also expect it to considerably speed up the generation of annotation sets for 3D and for deep-learning purposes.

## ACKNOWLEDGEMENTS

We are grateful to all crews during the different sampling cruises for their help and support. We would also like to thank D. Langenkämper and M. Zurowietz for their support

and the ongoing development of BIIGLE. We wish to express our appreciation to the editor and reviewers for their time and energy in providing insightful comments, which have greatly improved the manuscript. The French Oceanographic Fleet program provided the ship time.

### Funding
This work was supported by the Agence National pour la Recherche (ANR): project CERBERUS (contract number ANR-17-CE02-0003). The funders had no role in study design, data collection and analysis, decision to publish, or preparation of the manuscript.

### Grant Disclosures
The following grant information was disclosed by the authors:
Agence National pour la Recherche (ANR): ANR-17-CE02-0003.

### Competing Interests
The authors declare that they have no competing interests.

### Author Contributions
- Marin Marcillat conceived and designed the experiments, performed the experiments, analyzed the data, prepared figures and/or tables, authored or reviewed drafts of the article, and approved the final draft.
- Loic Van Audenhaege conceived and designed the experiments, performed the experiments, authored or reviewed drafts of the article, and approved the final draft.
- Catherine Borremans conceived and designed the experiments, authored or reviewed drafts of the article, and approved the final draft.
- Aurélien Arnaubec conceived and designed the experiments, authored or reviewed drafts of the article, and approved the final draft.
- Lenaick Menot conceived and designed the experiments, analyzed the data, authored or reviewed drafts of the article, and approved the final draft.

### Data Availability
The code is available at GitHub and Zenodo:

- https://github.com/marinmarcillat/CHUBACAPP.

- Marcillat Marin, Menot Lénaick, Van Audenhaege Loïc, & Borremans Catherine. (2023). Chubacapp, an open tool-box to process images, 3D models and annotation data from ROV and AUV (1.7.1). Zenodo. https://doi.org/10.5281/zenodo.7716971.

The images, navigation data and 3D reconstructions are available at: Marcillat Marin, Menot Lenaick, Van Audenhaege Loic, Borremans Catherine (2024). Example data for 2D image annotations onto 3D models. SEANOE. https://doi.org/10.17882/99108.

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
