# Peer review of "The best of two worlds: reprojecting 2D image annotations onto 3D models"

_PeerJ, doi:10.7717/peerj.17557_

## Round 0.1 · original submission · Major Revisions

In addition to the reviewers' comments, especially those referring to adding more detail to the methods, please pay also attention to the following:

1) I assume the geographic coordinates estimated by the navigation system of the vehicle are recorded in the exif header of the photographs, because Biigle can provide an approximate geolocation of annotations based the camera position, attitude (aim angles in xyz) and internal parameters (focal length etc). Please explain how Biigle estimates the distance from the camera to the real-life counterpart of the annotated image object. You do that by ray tracing, but unless Biigle uses the output of Mattise3D to provide that geolocation, it is unclear how the distance can be computed.

2) Seems to me Mattise#3D provides more than just one 3D model. You have a sparse point cloud, a dense point cloud, and a mesh. I suspect that most of the times, when you refer to the 3D model, it is the mesh what you mean. If so, please use the more precise term. Spell out what values of the user parameters were selected for the mesh creation. A mesh can be more or less detailed (from just being the outer envelope of the dense point cloud, to include holes and crevasses).

3) Spell out the coordinate system in which you made your measurements. WGS84 without any other qualifier is simply the longitude and latitude over the WGS84 ellipsoid, so distances in ‘WGS84’ are measured in decimal degrees, not meters. The length in meters of a degree of longitude depends on latitude, so there is no constant conversion factor. Since your measurements are in meters, it seems you may be using a projected coordinate system (maybe UTM?) that uses easting and northing instead of longitude and latitude, and which may still have WGS84 as datum. Please specify it. Please discuss also the implications of using the 3D model as geolocation reference. The 3D model averages the positioning errors of the vehicle navigation system, but it cannot cancel them fully, hence it will contain some degree of positional error. Does it matter if the remaining error is say 1m in the horizontal and 10 m in depth? Just to clarify, this remaining error is absolute, i.e., relative to the real-life position of the annotated objects, whereas what you are measuring is relative error, relative to the apparent position in the 3D model. If we could empty the oceans and survey your ground control objects with a centimetric accuracy GNSS instrument, chances are that the purported coordinates from the 3D model will differ by perhaps meters from those in the GNSS.

4) If I understood it well, you report error as the Euclidean distance between the projection into the 3D model of the corners of the annotation boxes drawn in the raw images and the corresponding corners of those drawn directly in the 3D model. Depending on their position relative to the center of the raw image, projected boxes may become quadrilaterals with no parallel sides when projected, hence, to base the accuracy in the corners seems not appropriate, as you probably draw the boxes in the 3D model as rectangles. Therefore, it would better to use the centroid of the boxes. And report horizontal and vertical accuracy separately.

5) As for the comment of the 2nd reviewer that there are commercial software that already allow for reprojection between raw images and geographic 3D space, this is relevant. I asked them to corroborate and they provided this link: https://www.agisoft.com/forum/index.php?topic=12781.0. Please explain how your workflow is different and please remove any claim of precedence. Your contribution is still relevant, as is open source, but it seems it is not the first.

Reviewer 1 ·

Basic reporting

In this study, the authors present a new method to reproject annotations from 2D image onto 3D models. While many tools are available to annotate 2D images, few software programs allow the annotation of 3D models. Moreover, the annotation process in 3D can be time consuming and, to date, cannot be carried out in a collaborative way (unlike 2D image annotation, which can be done using collaborative programs such as BIIGLE). The method presented here thus represents a real advantage, as it can leverage already existing annotations and put them in a 3D context, allowing the evaluation of species distribution in relation to terrain parameters at the centimeter scale.
Overall, with the exception of a few typos and grammatical errors, the manuscript is well written. However, more information is needed in the introduction and materials and methods to fully understand the importance of this study and how the analyses were carried out (see my detailed comments). In particular, the research question is well defined, but knowledge gaps filled by this study need to be a little clearer.
Nonetheless, I think this study is really valuable and will facilitate the annotation and analysis of large datasets in 3 dimensions.

Comments on Abstract
Lines 17 to 21 – When reading these sentences, readers may wonder why 3D models cannot be directly annotated: why is there a need to first annotate in 2D and then reproject onto a 3D model? The authors mention later in the Introduction that the resolution of 3D models can sometimes be too low to identify organisms, and no collaborative tools exist for 3D model annotation. Moreover, annotating large 3D models can require significant computing power. I think this should be briefly mentioned in the abstract as well to clearly justify the need to reproject 2D annotation onto 3D models.
Overall, the authors need to better show what the issue is and how this particular study is solving it.

Line 22 – Replace inherited by obtain?

Line 23 – Many readers may not be familiar with the term “bundle adjustment”. Please rephrase or briefly define this term for clarification.

Comments on Introduction
Line 36 – This sentence does not sound right to me, I would replace “while allowing” by “and allows”.

Line 38 – This is a little confusing, do you mean investigate spatial patterns at different scales?

Line 41 – Please replace “made in” by “from”.

Line 43 - "the" in front of fauna not needed.

Line 50 – Reference missing.

Line 65 – You can remove “objectives” in front of terrain metrics. You are not mentioning subjective metrics so it is not needed.

Line 75 to 78 – This is indeed an issue but you can always go back to the original images/video to identify the organisms and then add the annotation to the 3D model. The real issue here, is that going back to original images/video takes time. Therefore, your method represents a good alternative that will allow researchers to save significant amount of time. Moreover, I believe that you should insist more on the collaborative aspect, and the fact that there are now many available software programs for 2D image annotation that are widely used within the scientific community. Your method provides a good way to leverage data collected through these programs and put them in a 3D context.

Experimental design

The methods used in this study seem appropriate to address the research question. However, more details are needed in the Materials and Methods section to clarify some of the analyses.

Comments on Materials and Methods
Line 94 – Did you always use a downward-looking camera? How did you map vertical hydrothermal chimneys with a downward-looking camera?

Line 99 – I am confused by this sentence, it seems that there is a typo.

Line 100 – Reference missing.

Line 124 to 128 – I had to read this sentence a few times to fully understand it. I think if would be easier to understand if you clearly explained what disjoint images are, and how the function in Matisse 3D works. From what I understand this is a photomosaic built entirely based on raw navigation and altitude data (no optical matching feature included)? But then did you annotate all images or a certain number (percentage of total)? Did you annotate all visible fauna or only a few recognizable features? Which software program did you use for annotation? This should be clearer.

Lines 133 and 134 – How did you choose the 4 recognizable features? Was this visually random? Did you make sure that these features were of different sizes and located in different parts of the images (to cover any potential effect of image distortion)?

Line 135 – What is the csv report scheme? Mentioning that the spatial coordinates of all annotations were exported would be clearer.

Line 146 – Is ray tracing a commonly used method? Is there a reference you could cite here?

Line 147 – Is the location of the viewpoint determined by the camera characteristics? Is the ray perpendicular to the scene?
Which software program did you use for ray tracing. Was this all carried out in python? Were the reprojected annotation then added onto the 3Dmodel?

Line 161 – I believe “Evaluation of reprojection accuracy” would be a more precise and appropriate title.

Lines 163 and 164 – How were these distances measured? Which software program did you use?

Lines 166 to 169 – For clarification I would explicitly specify that these are annotation locations without reprojection.

Validity of the findings

All data have been provided and results support the authors’ conclusions. I have suggested additional discussion points, which I think are needed to fully understand the context of the study.

Comments on Results
Line 149 – Replace “allows highlighting” by “highlighted.

Lines 182 and 183 – Is there a way you could quantify overlap differences between images?
Also, as I mentioned earlier, from the method it was not clear at all that all these images, and the 3D model were annotated. Does that mean you annotated all fauna? Or only a few easily recognizable features to look at duplication percentage?

Line 192 – It should be specified that you are referring to Figure 4A.

Line 212 – Figure 4B.

Comments on Discussion
A lot of information is repeated from the introduction (especially in the first three paragraphs; e.g., regarding the lower resolution of 3D models, advantages of annotating in 2D vs 3D etc.). The Discussion should only bring new information, showing how your results are advancing the field.

Line 237 – What do you mean by variations? Interquartile range? Needs to be specified.

Lines 245 to 254 – A lot of this information is already in the Introduction. As I mentioned in earlier comments, it would be good to indicate in the Introduction that annotating 3D models takes a lot of time and, to date, cannot be done in a collaborative way, hence the importance of your study.

Line 255 – “Satisfactory” is a subjective term. You need to be more precise. for instance, you could say that locations of reprojected annotations are on average X times more accurate than without reprojecion.

Line 259 – How long does the reprojection process take? This should be indicated to really show that your method is allowing you to save significant time. Giving an order of magnitude of how much time is saved would be a good way to do that.

Lines 275 to 278 – This is already being done. Mapping platforms mounted on ROVs including LIDAR, multibeam sonar and stereo cameras have been developed to simultaneously map and image the seafloor at cm scales. These methods produce 3D models, bathymetry and photomosaics that fully overlap allowing the precise evaluation of species distribution in relation to topography. See Caress, D.W. et al., 2018. 1-cm Resolution Seafloor Surveys Combining Wide Swath Lidar, Multibeam Sonar, Stereo Cameras, and INS on an Articulating ROV Toolsled. AGU Fall Meeting OS33D-1920. For description of the method and Barry et al. Abyssal hydrothermal springs—Cryptic incubators for brooding octopus. Sci. Adv., 9 (34), eadg3247. DOI: 10.1126/sciadv.adg3247. for an example of how this method has been used.
With this mapping system reprojection is not needed, but not everyone has access to such technology and your method is definitely more cost-effective.
I think that all this should be discussed.

Line 279 – Do you mean this is a bottleneck in terms of time required to annotate all these images?

Additional comments

Comments on Figures and Tables
Some figures/Tables either do not have a title or legend. Both are needed to describe figures.

Figure 1 – Please add a legend describing the figure (what does it represent, what is the meaning of the different colors etc.).

Figure 2 – From which study site are these images from? Please add a scale.

Figure 3 – That would help if you specified that these are disjoint images (B).
I think it would also be good to visually point to a few duplicated features on the disjoint images to illustrate your point.

Figure 4 – This Figure does not have a title.
Please indicate that the y-axis is on a log scale. Also mention that A and B are represented on different scales, and that differences between both are thus greater than they appear to be at first glance.

Table 1 – Please add a legend.

·

Basic reporting

The authors describe well the advantages and challenges of the individual 3D and 2D processing approaches, and the motivation of their development. They give enough background on the state of the art. The structure is clear and the trail of though mostly good to follow. There are some minor dificits in the English language which are easily corrected (edited most of them), but some sections use vocabulary that does not fit well to serve their cause, and some paragraphs I failed to understand. I tried to suggest some clarifications in nomenclature.
The software and data are well referenced alhough the zenodo link to the toolsuite did not work for me. Github search did instead.

Experimental design

The research is within the scope of the journal as it has broad implications for habitat mapping. The goals and challenges are well defined. I have minor criticism regarding the treatment of acoustic navigation data as well as in the context of photogrammetric reconstruction accuracy. The approach taken to implement the workflow is flexible, timely and reproducible.

Validity of the findings

I cannot follow critical sections of the statistical analysis which may substantially suffer from wording issues. I suggest a revision involving a fluent English speaker. A table or supplement containing the actual measurements and statistics is not included and can thus not be evaluated or reproduced. The authors offer to make these available upon request, though.
The conclusions are aptly formulated although I do not share some minor implications stated therein.
I must point out that key elements of the described workflow have been an integral part of commercial photogrammetry software for a number of years, satisfying the primary objective of the workflow described here. In this context, the merit of this publication may lie in the offering of an open access, flexible alternative, while a major shortcoming is that it does not operate in real time.

Additional comments

This is a valuable and needed application of a recent solution to a community wide problem. It is important to the community to have such works of reference, even though they may address everyday workflow issues at first sight. I would like to suggest you expand a bit more on the quantitative description, individual challenges and typical features of your example data sets, elaborate a bit more on typical data structure and the state of the art in underwater navigation, streamline the figures to make them more intuitive, and consolidate your argumentation in the data analysis section.

---

## Round 0.2 · Minor Revisions

Dear Marin and co-authors,

I sent your revised manuscript to the same two reviewers who evaluated your original submission. They both are satisfied with the revisions you made and recommend it be accepted. I agree with them, but have entered the decision in the system as 'minor revisions' so that you can submit a final manuscript with a few minor revisions/comments I have inserted in the attached document (please disregard any comments on formatting; they were automatically added by my Word version). You don't need to create a full rebuttal letter, but I'd appreciate a short explanation of suggestions I made that you preferred not to follow. Congratulations!

Reviewer 1 ·

Basic reporting

This is the second time that I am reviewing this manuscript. The authors have addressed all my comments and I find that the quality of the manuscript has significantly improved and is now ready for publication. I have no additional comments. Congratulations to the authors on this great contribution.

Experimental design

No comment.

Validity of the findings

No comment.

·

Basic reporting

no comment

Experimental design

no comment

Validity of the findings

no comment

Additional comments

I am now satisfied with the many careful revisions done by the authors. No further comments. Well done and good luck!

---

## Round 0.3 · accepted · Accept

I've read the revised manuscript and the responses to my minor revisions and I believe the manuscript can be published in PeerJ. Regarding your question on papers using deep learning on 2D images, projecting the predicted boxes in 3D and removing duplicates, as said we are working on this topic in the realm of forests but unfortunately the manuscript is still in preparation, and I don't know of any paper exactly dealing with that topic. Congrats again!